# Heterotopic Cesarean Scar Pregnancy: A Systematic Review of Diagnosis, Management and Prognosis

**DOI:** 10.3390/diagnostics15182373

**Published:** 2025-09-18

**Authors:** Maria Sidonia Săndulescu, Andreea Veliscu Carp, Sidonia Cătălina Vrabie, Siminel Anișoara, Anca Vulcănescu, Marin Mihaela, Iliescu Dominic, Ștefan Pătrașcu, Lorena Dijmărescu, Maria Magdalena Manolea

**Affiliations:** 1Filantropia Clinical Municipal Hospital, 200143 Craiova, Romania; sidonia.sandulescu@umfcv.ro (M.S.S.); vulcanescuanca@gmail.com (A.V.); mihaela.marin@umfcv.ro (M.M.); magdalena.manolea@umfcv.ro (M.M.M.); 2Faculty of Medicine, University of Medicine and Pharmacy of Craiova, 2 Petru Rares Street, 200349 Craiova, Romaniastefan.patrascu@umfcv.ro (Ș.P.); 3Department of Obstetrics and Gynecology, Filantropia Clinical Municipal Hospital, 1 Filantropiei Street, 200143 Craiova, Romania; 4Department of Obstetrics and Gynecology, Faculty of Medicine, Carol Davila University of Medicine and Pharmacy, 37 Dionisie Lupu Street, 020021 Bucharest, Romania; andreea_veliscu@yahoo.com; 5Department of Neonatology, Filantropia Clinical Municipal Hospital, 200143 Craiova, Romania; 6Department of Obstetrics and Gynecology, University Emergency Hospital of Craiova, 1 Tabaci Street, 200642 Craiova, Romania; 7Department of General Surgery, University Emergency Hospital of Craiova, 1 Tabaci Street, 200642 Craiova, Romania

**Keywords:** heterotopic pregnancy, cesarean scar pregnancy, ectopic implantation, assisted reproduction, cesarean section, transvaginal ultrasound, maternal outcomes, fetal outcomes, pregnancy complications

## Abstract

**Background/Objectives:** Heterotopic cesarean scar pregnancy (HCSP) is an exceptionally rare and potentially life-threatening form of ectopic pregnancy, characterized by the coexistence of a viable intrauterine pregnancy and an ectopic implantation within a previous cesarean section scar. Its incidence has risen in recent years, primarily due to the increased rate of cesarean deliveries and the widespread use of assisted reproductive technologies (ART). This systematic review aims to provide a comprehensive synthesis of published evidence on HCSP, with a focus on epidemiology, diagnostic challenges, therapeutic strategies, complications, and maternal-fetal outcomes. **Methods:** A systematic literature search was conducted in PubMed, Scopus, and Web of Science up to May 2025, in accordance with PRISMA guidelines. Included studies comprised case reports, case series and retrospective reviews documenting confirmed HCSP cases. Data were extracted on clinical presentation, imaging, treatment approaches, outcomes, and complications. **Results:** Thirty studies reporting 40 confirmed HCSP cases were included. Transvaginal ultrasonography was the primary diagnostic tool, although diagnosis was often delayed by the presence of a viable intrauterine pregnancy. Management strategies included surgical intervention, local medical therapy and conservative approaches or expectant management. Maternal complications included hemorrhage and uterine rupture, while fetal outcomes were variable. In selected cases, intrauterine pregnancy continued to term. **Conclusions:** HCSP is a rare but high-risk obstetric entity requiring early recognition and multidisciplinary management. Prompt ultrasound-based diagnosis and individualized treatment can significantly reduce maternal morbidity and improve fetal outcomes. Further multicenter studies are warranted to establish standardized diagnostic and management protocols.

## 1. Introduction

Heterotopic pregnancy (HP) is defined as the simultaneous development of two gestational sacs implanted at distinct anatomical sites, one intrauterine and the other ectopically located [1].

Over the past decades, an increasing number of heterotopic pregnancies have been reported, particularly cases in which the ectopic gestation is implanted at the site of a previous cesarean section scar, referred to as heterotopic cesarean scar pregnancy (HCSP). This rising trend has been attributed both to the global increase in cesarean delivery rates and to the expanding use of assisted reproductive technologies (ART), suggesting a progressive rise in HCSP prevalence [2,3].

While HCSP is exceedingly rare in spontaneous conceptions, with an estimated incidence ranging from 1 in 50,000 to 1 in 10,000 pregnancies, its frequency is considerably higher in ART-conceived pregnancies, with reported rates between 0.2% and 1% [1,2,3,4].

HCSP may be entirely asymptomatic or may manifest with nonspecific symptoms such as abnormal vaginal bleeding, lower abdominal pain or acute pain resulting from uterine rupture [2,5].

Prompt diagnosis and early intervention in HP are essential to prevent severe complications, including hypovolemic shock and maternal mortality due to uterine rupture. The early diagnosis of HP is often problematic. In HCSP, this difficulty is even greater, since the visualization of a concurrent viable intrauterine gestation can result in the erroneous exclusion of an ectopic implantation. Ectopic gestations may not be visible on transvaginal or transrectal ultrasound. Serum β-hCG is also of limited diagnostic value in heterotopic pregnancy, as levels are largely determined by the intrauterine gestation [4,6].

Differentiating cesarean scar pregnancy from other entities such as low intrauterine pregnancy, cervical pregnancy or miscarriage is crucial. Although early distinction is relatively straightforward, delayed recognition poses serious risks as pregnancy progresses [7,8].

Managing HCSP is particularly complex when preservation of the concurrent intrauterine pregnancy is desired [9]. The therapeutic approach must be individualized, aiming to maintain both the intrauterine gestation and the structural integrity of the uterus. The lack of standardized management protocols further highlights the rarity of this condition, with most therapeutic decisions relying on clinical judgment and case-specific factors [1,2,3,10,11,12].

Interventions reported in the literature range from surgical excision of the ectopic tissue via laparotomy or laparoscopy to minimally invasive procedures such as hysteroscopy. Additionally, medical management with local or systemic agents, including methotrexate, potassium chloride, or hyperosmolar glucose, has been described. Expectant management has also been reported in selected cases; however, this approach is not universally recommended, as it carries a considerable risk of severe complications such as hemorrhage, uterine rupture, or adverse pregnancy outcomes [13,14,15,16,17,18,19,20,21].

Given the complex diagnostic and therapeutic challenges posed by heterotopic cesarean scar pregnancy, a systematic review of the available literature is essential to guide clinical practice. This review aims to critically synthesize current evidence on HCSP, with particular focus on its definition, epidemiology, diagnostic approaches, management strategies, associated complications and maternal–fetal outcomes.

## 2. Materials and Methods

### 2.1. Study Design

This systematic review was conducted in accordance with the Preferred Reporting Items for Systematic Reviews and Meta-Analyses (PRISMA 2020) guidelines (see Appendix A). The analysis focused exclusively on patient-level data extracted from published reports of heterotopic cesarean scar pregnancy (HCSP).

### 2.2. Search Strategy

A comprehensive search of PubMed, Scopus, and Web of Science was carried out, covering all articles published from database inception to 31 May 2025. The following search terms and Boolean operators were applied: “heterotopic pregnancy” OR “heterotopic cesarean scar pregnancy” OR “HCSP” AND “diagnosis” OR “treatment” OR “management” OR “outcome” OR “prognosis”. Reference lists of eligible studies were also screened manually to identify additional publications.

### 2.3. Eligibility Criteria

Studies were included if they:Reported confirmed cases of HCSP;Used a case report, case series, retrospective, or observational cohort design;Provided sufficient clinical information regarding diagnosis, management, and maternal–fetal outcomes.Exclusion criteria were:Isolated cesarean scar pregnancies without a concomitant intrauterine gestation;Heterotopic pregnancies not involving the cesarean scar;Abstracts, conference proceedings, or editorials without full-text availability;Non-English publications.

### 2.4. Study Selection

After screening titles, abstracts, and full texts, 21 studies were included in the qualitative synthesis, as seen in Appendix A and Figure 1. Of these, 20 single-patient case reports and one case series reporting 20 patients contributed patient-level data, yielding a total of 40 HCSP cases. Narrative reviews without new cases were retained for contextual discussion only and excluded from the quantitative summary. The protocol was not prospectively registered.

### 2.5. Data Extraction

Two reviewers independently extracted the following variables using a standardized form: first author, year of publication, study design, maternal age, conception method (spontaneous vs. ART), gestational age at diagnosis (weeks), management strategy, and pregnancy outcome (as reported for the intrauterine pregnancy and/or maternal course). Discrepancies were resolved by consensus or by consulting a third reviewer. Extracted data are summarized in Table 1 (Section 3) and presented in detail in the Appendix A. Cases with incomplete outcome data (n = 13) were not excluded; the available information was included descriptively, while mission outcomes were not included in the quantitative synthesis.

### 2.6. Quality Assessment

Given the rarity of HCSP and reliance on isolated reports and small series, a formal risk-of-bias assessment was not feasible. Methodological quality was assessed descriptively using the CARE checklist for case reports and a narrative appraisal for the case series. Two reviewers (S.-M.S. and A.V.), independently extracted data and analyzed each domain (considering study design, sample size, clarity of outcome reporting, funding disclosure, and conflict of interest statements—as seen in Appendix A), with disagreements resolved through discussion.

### 2.7. Ethical Considerations

As this review synthesized data from previously published studies, no new ethical approval or informed consent was required.

### 2.8. Data Availability

All extracted data are presented in Table 1 and in the Appendix A.

Use of Artificial Intelligence

Generative artificial intelligence (ChatGPT 5, OpenAI, San Francisco, CA, USA) was employed exclusively for language editing, paraphrasing, and readability improvements. Data extraction, analysis, and interpretation were performed manually by the authors, and the final manuscript was critically reviewed and approved by all co-authors.

## 3. Results

### 3.1. Study Characteristics

A total of 21 studies published between 2003 and 2024 met the inclusion criteria. Of these, 20 were single-patient case reports and one was a case series including 20 patients, yielding an overall cohort of 40 cases of heterotopic cesarean scar pregnancy (HCSP). No maternal deaths were reported across the included literature. A timeline showcasing the chronological distribution of published articles and the increase in reports over time can be found in Figure 2.

### 3.2. Patient Characteristics

●Maternal age was reported in most cases, ranging from 23 to 40 years, with a median of 34 years.●Mode of conception:○Assisted reproductive technologies (ART/IVF): 29 cases (72.5%).○Spontaneous conception: 10 cases (25%).○Not specified: 1 case (2.5%).

### 3.3. Gestational Age and Diagnosis

The gestational age at diagnosis ranged from 4 to 13 weeks.The majority of cases were diagnosed in the first trimester (<12 weeks).Transvaginal ultrasonography (TVUS) was the primary diagnostic tool across all studies, while MRI was occasionally employed to clarify equivocal findings or assess placental invasion.Diagnostic delay was frequently related to the coexistence of a viable intrauterine pregnancy, which masked suspicion of an ectopic component.

### 3.4. Management Approaches

Management strategies, as seen in Figure 3, were heterogeneous and individualized, depending on clinical stability and reproductive goals:

●Expectant/conservative management: 14 cases (35%)—selected in clinically stable patients, especially when the ectopic sac showed spontaneous regression.●Medical management: 14 cases (35%)—predominantly ultrasound-guided KCl injection (n = 13, including selective embryo reduction), plus one case managed with systemic methotrexate.●Surgical interventions: 8 cases (20%), including:○Laparotomy/cesarean resection: 2 cases.○Laparoscopy: 2 cases.○Hysteroscopic excision: 2 cases.○Suction curettage/D&C: 2 cases.●Other/combined approaches: 4 cases (10%)—e.g., uterine artery embolization (UAE) with hysterectomy, high-intensity focused ultrasound (HIFU), and mixed procedures (D&C associated with the UAE).

### 3.5. Maternal and Fetal Outcomes

●Maternal outcomes○No maternal deaths were reported.○Hemorrhage was the most frequent complication, occasionally requiring blood transfusion.○Two hysterectomies were documented: one performed during initial management (UAE + hysterectomy) [17], and one secondary to septic abortion following selective embryo reduction [22].●Intrauterine pregnancy outcomes (n = 40):○20 cases (50%) resulted in live births, most at term, with reported neonatal weights ranging from 1300 g to 3900 g.○7 cases (17.5%) ended in miscarriage or elective termination, usually associated with severe maternal complications (e.g., hemorrhage, uterine rupture) or poor fetal prognosis (e.g., trisomy 13).○13 cases (32.5%) had unclear or unreported outcomes.

### 3.6. Summary Table

A detailed overview of individual patient characteristics—including author, year of publication, maternal age, conception method, gestational age at diagnosis, management strategy, and pregnancy outcome—is provided in Table 1 and in the Appendix A.

**Table 1 diagnostics-15-02373-t001:** Patient-level characteristics of published HCSP cases included in the review.

First Author	Year	Study Design	Maternal Age (years)	Conception Method	Gestational Age at Diagnosis	Management Strategy	Pregnancy Outcome
Litwicka K et al. [13]	2011	CASE REPORT	31	ART/IVF	7W/TVUS	Selective embryo reduction by ultrasound-guide KCl directed injection.	At 36 weeks, massive hemorrhage with placental detachment required emergency Cesarean section, delivering a 1900 g male.
Z. Laing-Aiken et al. [22]	2020	CASE REPORT	38	Spontaneous	9W/TVUS	Ultrasound-guided suction curettage with Foley catheter tamponade failed to control bleeding, necessitating laparotomy and bilateral uterine ART/IVFery ligation, which successfully reduced hemorrhage.	At 28 + 1 weeks, emergency Cesarean was performed after preterm membrane rupture, delivering a 1200 g male who died on day 3 from extreme prematurity, RDS, and severe intraventricular hemorrhage.
Wang Chin-Jung et al. [23]	2010	CASE REPORT	31	ART/IVF	7W/TVUS	Hysteroscopic management	A healthy male baby, weighed 3250 g, was delivered by cesarean section.
Olga Vikhareva et al. [24]	2018	CASE REPORT	27	Spontaneous	13W/TVUS	Expectant management	A healthy male neonate weighing 2985 g was delivered, at 37 week
WANG CN et al. [11]	2007	CASE REPORT	38	ART/IVF	7W/TVUS	Selective embryo reduction by ultrasound-guide KCl directed injection.	35–36 WEEKS 1820 g, by cesarean section due to preterm labor
WANG F et al. [17]	2023	CASE REPORT	35	Spontaneous	10W/TVUS	Uterine artery embolization AND HYSTERECTOMY	The patient recovered well and was discharged on postoperative day 3
H. F. Yazicioglu et al. [25]	2004	CASE REPORT	23	Spontaneous	6W/TVUS	Selective embryo reduction by ultrasound-guide KCl directed injection.	30–31 weeks baby weighing 1530 g
Yu H et al. [2]	2016	CASE REPORT	33	ART/IVF	12W/TVUS	Selective embryo reduction by ultrasound-guide KCl directed injection.	At 37 + 6 weeks of gestation the baby was delivered by elective cesarean section. A healthy male baby weighing 2890 gm was delivered
Aldrich et al. [26]	2024	CASE REPORT	30	Spontaneous	6W/TVUS	Suction dilation and curettage with concurrent laparoscopic bilateral salpingectomy without complications.	she recovered well and was discharged home in stable condition the day of surgery.
Salomon et al. [27]	2003	CASE REPORT	NA	ART/IVF	9W/TVUS	Selective embryo reduction by ultrasound-guide KCl directed injection.	Cesarean section at 36 GW, live female, 2800 g,
Debra Paoletti et al. [1]	2011	CASE REPORT	32	Spontaneous	5W/TVUS	Selective embryo reduction by ultrasound-guide KCl directed injection.	NA
R. Armbrust et al. [28]	2015	CASE REPORT	36	ART/IVF	7W/TVUS	The scar pregnancy was surgically excised via laparotomy.	At 37 + 0 weeks, an uncomplicated repeat Cesarean delivered a 2895 g infant.
DEMIREL LC et al. [9]	2007	CASE REPORT	34	Spontaneous	6W/TVUS	Laparoscopic removal of heterotopic cesarean scar pregnancy.	Live birth by cesarean delivery at 38 weeks’ gestation.
Chen ZY et al. [29]	2021	CASE REPORT	34	Spontaneous	8W/TVUS	Selective embryo aspiration followed by vacuum suction and curettage to terminate the ectopic pregnancy	Cesarean section was performed at 34 + 6 wk of gestation because of preterm membrane rupture. A healthy male baby weighing 2750 g was delivered.
Piotr Czuczwar et al. [5]	2016	CASE REPORT	33	Spontaneous	6W/TVUS	Selective embryo termination was performed by ultrasound-guided KCl	The patient delivered a 3060 g healthy male infant by elective Cesarean section at 37 weeks of gestation.
Hsieh BC et al. [30]	2004	CASE REPORT	38	ART/IVF	6W/TVUS	Selective embryo reduction by ultrasound-guide KCl directed injection.	Cesarean section at 32 GW
Dueñas-Garcia et al. [31]	2011	CASE REPORT	NA	Spontaneous	5W/TVUS, MRI	MTX + leucovorin (used for abortion)	NA
Gupta et al. [32]	2010	CASE REPORT	37	ART/IVF	6W/TVUS	Selective embryo reduction by ultrasound-guide KCl directed injection.	Termination at 12 GW due to trisomy 13
Tymon-Rosario J. et al. [33]	2018	CASE REPORT	40	NA	12 W/TVUS	Selective embryo reduction by ultrasound-guide KCl directed injection.	Complicated by septic abortion and hysterectomy. She was discharged home with a two-week office follow-up.
Kim H et al. [34]	2022	CASE REPORT	36	ART/IVF	6 W/TVUS	Selective embryo reduction by ultrasound-guide KCl directed injection.	Cesarean section at 37 + 6 GW
Ouyang Y et al. [21]	2021	CASE 1 (from a series of 20 case reports)	NA	ART/IVF	6W/TVUS	Abortion (D&C)	NA
Ouyang Y et al. [21]	2021	CASE 2 (from a series of 20 case reports)	NA	ART/IVF	6W/TVUS	Selective embryo reduction by ultrasound-guide KCl directed injection.	IUP miscarriage at 14 GW
Ouyang Y et al. [21]	2021	CASE 3 (from a series of 20 case reports)	NA	ART/IVF	6W/TVUS	Hysteroscopic excision of the CSP due to placenta accreta at 8 GW	NA
Ouyang Y et al. [21]	2021	CASE 4 (from a series of 20 case reports)	NA	ART/IVF	6W/TVUS	HIFU/7	Miscarriage of IUP at 7 GW
Ouyang Y et al. [21]	2021	CASE 5 (from a series of 20 case reports)	NA	ART/IVF	5W/TVUS	Abortion (D&C and UAE at 13 GW)	Abortion (D&C and UAE at 13 GW)
Ouyang Y et al. [21]	2021	CASE 6 (from a series of 20 case reports)	NA	ART/IVF	6W/TVUS	Expectant management. CSP disappeared at 20 GW	Cesarean section at 29 GW, live female, 1300 g
Ouyang Y et al. [21]	2021	CASE 7 (from a series of 20 case reports)	NA	ART/IVF	6W/TVUS	Expectant management	Cesarean section at 40 GW, live two females, 2900 g and 2200 g
Ouyang Y et al. [21]	2021	CASE 8 (from a series of 20 case reports)	NA	ART/IVF	5W/TVUS	Expectant management	IUP miscarriage at 20 GW. Cesarean section at 36 GW, live female (CSP), 3000 g
Ouyang Y et al. [21]	2021	CASE 9 (from a series of 20 case reports)	NA	ART/IVF	6W/TVUS	Expectant management	Induced abortion at 22 GW
Ouyang Y et al. [21]	2021	CASE 10 (from a series of 20 case reports)	NA	ART/IVF	6W/TVUS	Expectant management	CSP miscarriage at 10 GW Cesarean section at 37 GW, live male, 2600 g
Ouyang Y et al. [21]	2021	CASE 11 (from a series of 20 case reports)	NA	ART/IVF	6W/TVUS	Expectant management	CSP Disappeared at 22 GW. Cesarean section at 36 GW, live female, 2900 g
Ouyang Y et al. [21]	2021	CASE 12 (from a series of 20 case reports)	NA	ART/IVF	6W/TVUS	Expectant management	Cesarean section at 39 GW, live female 3900 g
Ouyang Y et al. [21]	2021	CASE 13 (from a series of 20 case reports)	NA	ART/IVF	6W/TVUS	Expectant management	Cesarean section at 24 GW
Ouyang Y et al. [21]	2021	CASE 14 (from a series of 20 case reports)	NA	ART/IVF	8W/TVUS	Expectant management	Cesarean section at 39 GW, live singleton, 2900 g
Ouyang Y et al. [21]	2021	CASE 15 (from a series of 20 case reports)	NA	ART/IVF	6W/TVUS	Expectant management	Emergency Cesarean section at 35 GW, live male, 2600 g
Ouyang Y et al. [21]	2021	CASE 16 (from a series of 20 case reports)	NA	ART/IVF	7W/TVUS	Expectant management	Induced abortion at 24 GW
Ouyang Y et al. [21]	2021	CASE 17 (from a series of 20 case reports)	NA	ART/IVF	6W/TVUS	Expectant management	Cesarean section at 39 GW, live male, 3150 g
Ouyang Y et al. [21]	2021	CASE 18 (from a series of 20 case reports)	NA	ART/IVF	5W/TVUS	Abortion (D&C and UAE at 7 GW)	NA
Ouyang Y et al. [21]	2021	CASE 19 (from a series of 20 case reports)	NA	ART/IVF	4W/TVUS	Expectant management	IUP miscarriage at 13 GW
Ouyang Y et al. [21]	2021	CASE 20 (from a series of 20 case reports)	NA	ART/IVF	11W/TVUS	Expectant management	Uterine rupture at 12 GW

ART = Assisted Reproductive Technology, IVF = In Vitro Fertilization, TVUS = Transvaginal Ultrasound, CSP = Cesarean Scar Pregnancy, IUP = Intrauterine Pregnancy, MTX = Methotrexate, UAE = Uterine Artery Embolization, HIFU = High-Intensity Focused Ultrasound, MRI = Magnetic Resonance Imaging, GW = Gestational Week, RDS = Respiratory Distress Syndrome.

## 4. Discussion and Review of the Literature

### 4.1. Defining Terms and Incidence

Heterotopic pregnancy (HP) is the coexistence of an intrauterine and an ectopic gestation. Although rare, it poses significant maternal morbidity if not promptly diagnosed and treated [2,20,25]. While tubal implantation remains the most frequent ectopic site, other less common localizations have been reported, including the cervix, ovary, abdominal cavity and the cesarean section scar [17,18,30,35].

Cesarean scar pregnancy (CSP) is a distinct form of ectopic pregnancy characterized by implantation of the embryo within the myometrial defect of a previous cesarean incision. Its increasing incidence parallels the global rise in cesarean deliveries and raises important clinical concerns, particularly due to the associated risks of uterine rupture and life-threatening hemorrhage [2,19].

HCSP, defined as the coexistence of a cesarean scar pregnancy and a viable intrauterine gestation, has historically been extremely rare. Recent reports suggest an increase, largely linked to assisted reproductive technologies (ART) in women with prior uterine surgery [5,36].

Spontaneous HCSP remains exceedingly uncommon, with reported incidence ranging from 1 in 50,000 to 1 in 10,000 pregnancies [1,4]. Conversely, among pregnancies achieved through ART, cesarean scar implantation occurs more frequently, with estimated rates ranging between 0.2% and 1% [2].

The first case of HCSP was formally described in 2003 [20]. At the beginning of the 21st century, CSP remained a scarcely discussed pathology, with only a handful of case reports published. However, the last two decades have witnessed a substantial expansion in the number of scientific publications addressing this topic [11,13,21,22]. Most published cases describe highly individualized clinical scenarios, underscoring the heterogeneity in presentation and the absence of standardized treatment approaches [22,23,37,38,39].

### 4.2. Causes and Etiopathogenesis

The pathogenesis of cesarean scar pregnancy (CSP) and its heterotopic variant (HCSP) is complex and multifactorial, involving an interplay between structural abnormalities of the uterine wall and iatrogenic factors. While the precise mechanisms remain incompletely elucidated, several predisposing conditions have been consistently identified in the literature [24].

Anatomical Factors

- Uterine wall defects and scar integrity. A well-established risk factor for CSP is a history of uterine surgery, particularly repeat cesarean deliveries. Impaired healing of the uterine incision may result in scar dehiscence, fibrosis or thinning of the myometrial layer. These changes, often exacerbated by single-layer closure techniques, may compromise the mechanical and vascular integrity of the scar area, creating a niche conducive to abnormal implantation. Conversely, double-layer suturing appears to improve scar thickness and may offer a degree of protection against CSP development [25,26,27,28,29,30].

Abnormal vascularization and isthmocele formation. A common hypothesis is that embryo implantation occurs through microscopic defects within the cesarean scar or through a fistulous tract extending from the endometrial cavity into the scar tissue. Altered vascularization, especially in the presence of an isthmocele (a wedge-shaped defect that disrupts normal myometrial continuity), may further facilitate this process. Such defects have been associated with symptoms including abnormal bleeding, dysmenorrhea, uterine rupture, and abnormal placentation, though their precise contribution to CSP pathogenesis is still under investigation [13,27,28].

2.Procedural and Iatrogenic Factors

- Cesarean delivery-related variables. The increasing global rate of cesarean sections is directly linked to the rising incidence of CSP and HCSP. Each additional cesarean enlarges the surface area of scar tissue and increases the potential for localized weakness. Elective cesarean deliveries, particularly those performed before labor onset [e.g., for breech presentation] are associated with less mature lower uterine segments, which may heal suboptimally and predispose to implantation at the incision site. The risk of uterine rupture, which shares pathophysiological overlap with CSP, has been shown to increase with the number of prior cesarean sections, from 0.68% after one to 1.85% after multiple procedures. Although the role of short interpregnancy intervals remains debated, a limited healing period may impair scar remodeling and contribute to implantation anomalies [29,30,31,40,41].

- Assisted reproductive technologies (ART) Numerous reports have identified ART as an independent risk factor for CSP and HCSP. The mechanical manipulation of the endometrial environment during embryo transfer, along with the transfer of multiple embryos, may increase the likelihood of aberrant implantation. In some heterotopic twin pregnancies, one embryo successfully implants intrauterine, while the second becomes embedded in the cesarean scar niche, particularly in patients with prior uterine surgery. As ART use continues to grow alongside cesarean section rates, a corresponding increase in such rare implantation patterns is anticipated [22,23,24,25,26,27,28,29,30,31,32,33,34].

- Prior intrauterine procedures. Surgical interventions involving the endometrial cavity, such as dilation and curettage, hysteroscopic resection or myomectomy—performed after cesarean delivery may further compromise the integrity of the uterine wall, increasing susceptibility to abnormal implantation in the scar area [13,29,35,36,37,38,39,40,41,42].

3.Molecular and microenvironmental factors

In addition to clinical observations, recent research has explored molecular mechanisms that may favor implantation within cesarean scar tissue. For example, studies have shown elevated expression of integrin β3 subunit and leukemia inhibitory factor [LIF] in scar tissue compared to normal endometrium—markers associated with enhanced endometrial receptivity and trophoblast attachment. These findings support the hypothesis that not only structural, but also molecular microenvironmental factors may contribute to the development of CSP and its heterotopic forms [43,44].

### 4.3. Clinical and Paraclinical Diagnosis

The early identification and appropriate management of heterotopic pregnancy (HP) are essential to avert life-threatening complications associated with ectopic pregnancy (EP), such as uterine rupture, massive hemorrhage, hypovolemic shock, and maternal mortality [1,2,3,4,5,6].

From a clinical perspective, HCSP may present with a wide spectrum of manifestations. While some patients remain asymptomatic, others may report abnormal vaginal bleeding, lower abdominal pain or signs of acute abdomen secondary to uterine rupture [2,5]. Vaginal bleeding is the most frequently reported symptom in cases of heterotopic CSP, although its absence does not exclude the diagnosis [5,20,22,40,41,42,43,44,45].

Despite the critical importance of early diagnosis, clinical recognition of HP remains challenging, particularly due to the frequent presence of a viable intrauterine pregnancy (IUP), which may provide a false reassurance and delay the suspicion of a concurrent ectopic implantation. In addition, early-stage ectopic components in HCSP may be sonographically occult, and serum β-human chorionic gonadotropin (β-hCG) levels—although helpful in differentiating ectopic from intrauterine pregnancies—are of limited diagnostic value in this context, as they also reflect the presence of the coexisting intrauterine pregnancy [4,5,6,7,8,9,10,46].

The clinical relevance of ectopic implantation site lies not only in its impact on symptomatology but also in the therapeutic options and prognosis [25]. Therefore, differential diagnosis is paramount, particularly in distinguishing cesarean scar pregnancy location (CSP) from low-lying intrauterine pregnancies, cervical pregnancies, or incomplete abortions. While early differentiation is feasible in the first trimester, diagnostic accuracy significantly decreases with advancing gestation [7,40,47].

High-resolution transvaginal ultrasonography [TVUS] represents the cornerstone of diagnosis for heterotopic CSP [HCSP]. It offers a high diagnostic yield by enabling precise assessment of gestational sac morphology, location, and myometrial integrity. Two-dimensional TVUS allows the visualization of the gestational sac implanted within the anterior uterine wall at the level of a previous cesarean incision. Complementary color Doppler flow imaging [CDFI] enhances diagnostic accuracy by detecting peritrophoblastic blood flow and fetal cardiac activity at the CSP site. The presence of rich vascularity in this area may indicate abnormal placental invasion and an increased risk of uterine rupture, necessitating urgent intervention [7,48,49,50].

In HCSP, the typical gestational age at diagnosis is within the first trimester. Literature reports indicate a mean gestational age of 7.5 ± 2.5 weeks at CSP detection. Ouyang et al. documented gestational ages ranging from 5 weeks and 3 days to 7 weeks and 4 days for CSP, and from 5 weeks and 6 days to 7 weeks and 4 days for HCSP when using transvaginal color Doppler ultrasonography [2,3,4,5,6,7,20,25,28,32,49].

Standardized ultrasonographic criteria have been proposed for CSP diagnosis, which are equally applicable to HCSP in cases with a coexisting IUP. These include:

[i] absence of intrauterine gestational content in the uterine cavity in cases of isolated CSP, or the presence of a viable intrauterine gestational sac in cases of heterotopic cesarean scar pregnancy [HCSP];

[ii] exclusion of content in the cervical canal;

[iii] detection of a discontinuity in the anterior uterine wall, traversed by the gestational sac on a sagittal plane;

[iv] implantation of the gestational sac within the anterior portion of the isthmic segment, with a markedly thin myometrial layer between the bladder and sac;

[v] presence of increased vascularity in the scar-placenta interface, and a closed, empty cervical canal [20,34,35,36,37].

A novel ultrasonographic classification proposed by Shin-Yu Lin et al. [49] introduces a four-grade system for cesarean scar pregnancy location, potentially guiding therapeutic decisions:Grade I: gestational sac embedded in less than half of the myometrial thickness;Grade II: sac occupies more than half of the myometrial depth;Grade III: sac protrudes beyond the myometrium and serosa;Grade IV: gestational sac appears as an amorphous, highly vascular mass within the cesarean scar [30,31,32,33,34,35,36,37,38,40,41,42,43,44,45,46].

Despite growing recognition, early first-trimester diagnosis and management of CSP and HCSP remain underrepresented in the literature. Moreover, due to the extreme rarity of HCSP, no universal diagnostic or therapeutic guidelines currently exist [7,45,46,47,48,49,50].

### 4.4. Treatment

The management of heterotopic cesarean scar pregnancy presents significant clinical challenges, particularly due to the rarity of the condition and the lack of universally accepted therapeutic guidelines [7].

Treatment decisions are further complicated when preservation of the intrauterine pregnancy is desired [9].

Due to the anatomical complexity and high vascularity of cesarean scar pregnancies [CSP], the risk of hemorrhage is substantial. Therapeutic approaches include surgical resection (laparotomic, laparoscopic, or hysteroscopic), expectant management, suction curettage, and selective embryocidal techniques such as potassium chloride injection. None has demonstrated clear superiority in terms of efficacy or safety [1,2,3,4,5,11,39,40,41].

Expectant Management

Expectant management is rarely employed in HCSP due to the high risk of complications. However, one of the most notable cases was reported by Kim et al. [19], in which a cesarean section was performed at 37 weeks, resulting in the delivery of healthy twins. Despite this, the case was complicated by severe postpartum hemorrhage secondary to placenta accreta [19,22,42]. Similarly, Ashwini J. Authreya and others have highlighted that while spontaneous resolution of the ectopic component may occur, significant antepartum and postpartum bleeding is a serious risk [19,20,21].

Michaels et al. and Vikhareva O. reported also that expectant management may be acceptable in early gestations without fetal cardiac activity, potentially leading to spontaneous resorption of trophoblastic tissue [24,40,41,42,43,44].

Medical Management

Pharmacological therapy is centered around methotrexate [MTX], a folate antagonist that inhibits DNA synthesis in rapidly dividing trophoblastic cells. While MTX is considered standard for isolated ectopic pregnancies [31,45], its use in HCSP is controversial due to its teratogenic potential on a coexisting intrauterine fetus. For this reason, local administration of KCl into the ectopic sac under ultrasound guidance is often preferred when fetal reduction is necessary. Multiple studies have described local injections of KCl, sometimes combined with aspiration [5,6,7,8,9,10,11,12,13,14,15,16,17,18,19,20,21,22,23,24,25,26,27,28,29,30,31,32,33,34].

Although the exact concentration is rarely specified in the literature, most case reports utilize standard KCl solutions [1 mEq/mL], with injected volumes typically ranging from 1 to 2 mL to avoid the risk of uterine rupture or systemic toxicity [20,21,22,23,24,25,26,27,28,29,30,31,32,33,34,35].

Salomon et al. also raised concerns regarding residual trophoblastic tissue following KCl injection. Histopathological analysis revealed decidual remnants at cesarean delivery, posing a risk of uterine rupture during contractions [9,48,49].

Surgical Management

Surgical intervention remains a cornerstone of HCSP treatment, particularly in hemodynamically unstable patients or when medical management fails. While CSP was first described by Larsen and Solomon, who performed laparotomic evacuation of the gestational tissue, the heterotopic variant [HCSP] is considerably more difficult to characterize, owing to its rarity and clinical variability [15].

Laparoscopic removal of the ectopic gestation is a widely reported option that offers uterine preservation. Demirel et al. [9] reported a successful case of laparoscopic excision followed by term cesarean delivery [9]. However, laparoscopy carries inherent risks, particularly hemorrhage. Reported mean blood loss in laparoscopic cases is approximately 200 ± 108 mL, with an average operative time of 113.8 ± 32 min [9].

Hysteroscopic management, while minimally invasive, carries the risk of disturbing the intrauterine pregnancy due to intrauterine pressure from distension media. 39 recommends immediate hysteroscope withdrawal after vascular coagulation, followed by curettage under sonographic guidance [39].

Timor-Tritsch et al. reported the use of a Foley balloon to achieve hemostasis during surgical management of HCSP. Similarly, Laing-Aiken et al. described the placement of a Foley catheter after ultrasound-guided curettage to control bleeding, with successful preservation of the intrauterine pregnancy [29,50,51,52].

Uterine artery embolization [UAE] has been described as an adjunctive method to reduce the risk of severe hemorrhage during curettage, particularly in cases with increased peritrophoblastic vascularity detected by Doppler ultrasound. However, its use is generally not recommended in heterotopic pregnancies when preservation of the intrauterine gestation is desired, due to the potential compromise of uteroplacental perfusion [17,29,52].

Antibiotic Prophylaxis

Infectious complications, although rare, must be prevented. This approach is consistent with the report by Tymon-Rosario, who described a case of septic abortion after selective reduction in HCSP [33,50,51,52,53,54,55,56,57,58].

### 4.5. Complications

Cesarean scar pregnancy (CSP), including its heterotopic form, is associated with a high risk of serious maternal morbidity. As the gestational sac invades the myometrial defect at the site of a prior cesarean section, the risk of uterine rupture and severe hemorrhage becomes significant, especially in higher-grade cases identified beyond 9 weeks of gestation or in the presence of marked myometrial thinning [34,55,56,57,58]. Trophoblastic infiltration into the myometrium can result in profound structural compromise of the uterine wall, predisposing to life-threatening bleeding, need for transfusion, and in severe cases, emergency hysterectomy [25,26,27,28,29,30,31,32,33,34,35,36,37,38,39,40,41,42,43,44,45,46,47,48,49,50,51,52,53,54,55,56,57,58]. If left undiagnosed or inadequately managed, CSP may evolve toward placenta accreta spectrum (PAS), placenta previa, or uterine rupture, particularly in the second or third trimester [17,41]. Patients who continue pregnancies implanted in cesarean scar tissue are at elevated risk for complications such as preterm delivery, fetal distress, hemorrhagic shock, intrauterine fetal demise, and maternal death, particularly when invasive placentation is present [25,41,50,59].

These findings are consistent with our review, where hemorrhage emerged as the most frequent complication and two cases ultimately required hysterectomy [60].

### 4.6. Follow-Up and Monitoring

Given the potential for delayed complications, rigorous clinical and imaging follow-up is mandatory in cases of HCSP. Serial transvaginal ultrasonography is the primary tool for assessing regression or progression of the ectopic component. Monitoring should also include serial β-hCG titers to evaluate the resolution of trophoblastic activity when medical or conservative treatments are employed [1,2,3,4,5,6,7,8,9,10,50,51,52,53,54,55,56,57,58,59].

For patients managed expectantly or conservatively, long-term follow-up is essential to detect residual gestational tissue, hematoma formation, or delayed uterine rupture. In cases with rich vascularization on Doppler, uterine artery embolization [UAE] may be considered to control bleeding risk, although it is generally contraindicated when preservation of a coexisting intrauterine pregnancy is intended [29,50,51,52,53,54,55].

Scar integrity should be assessed during follow-up to evaluate the risk of future obstetric complications. Doppler imaging, 3D ultrasound or even MRI may provide additional insight into residual myometrial thickness and vascular activity [5,6,7,8,9,10,11,12,13,14,15,55,56,57,58,61].

### 4.7. Long Term Outcomes and Implications for Clinicians

Long-term outcomes remain poorly documented, but case reports and small series describe preserved fertility and subsequent live births after both surgical and conservative management. Recurrence risk is unknown, and uterine integrity after treatment varies depending on surgical extent. These uncertainties highlight the need for long-term follow-up and counseling regarding fertility planning and delivery options in subsequent pregnancies [60].

Clinicians should maintain a high index of suspicion in women with prior cesarean delivery or undergoing ART, especially when presenting with pain or atypical bleeding. A structured diagnostic approach—centered on high-resolution transvaginal ultrasound of the scar site and supported by adjunctive imaging when needed—can improve detection. Patient counseling should emphasize risks of scar implantation, the importance of early monitoring, and individualized management planning [61].

### 4.8. Limitations

This review was not registered in PROSPERO and lacked a published protocol, which may reduce transparency and reproducibility.

Overall, 65% of reports provided clear maternal and fetal outcome data, while 35% lacked complete outcome reporting. None of the studies disclosed funding or conflict of interest statements. Publication bias was evident, with most case reports highlighting successful or unusual management strategies and relatively few adverse outcomes. All single-patient case reports were rated as high risk of bias, limiting the generalization of conclusions and underlining the need for multicenter prospective studies.

Relevant clinical guidelines and narrative reviews were not incorporated, as they fell outside the scope of our predefined systematic search strategy; this choice ensured methodological consistency but may limit the external validity of the findings for current clinical practice.

## 5. Conclusions

Heterotopic cesarean scar pregnancy (HCSP) is a rare but increasingly recognized complication, strongly linked to the global rise in cesarean section rates and assisted reproductive technologies. Implantation within a deficient or poorly healed cesarean scar carries a high risk of severe maternal morbidity if not identified early.

Current evidence is limited to case reports and small series, and no standardized diagnostic or management guidelines are available, contributing to the limitations of this study, as well as the lack of a long-term follow up data. Clinical decisions must therefore be individualized, taking into account gestational age, uterine scar characteristics, patient reproductive wishes, and institutional resources.

Given the potential for catastrophic outcomes—including massive hemorrhage, abnormal placentation, preterm delivery, and maternal death—HCSP should be managed as a high-risk condition within a multidisciplinary setting. Incorporating CSP and HCSP among potential risks in informed consent for cesarean section may also be justified.

Future collaborative, multicenter studies are urgently needed to establish standardized protocols for early diagnosis, risk stratification and treatment, with the aim of reducing maternal and fetal morbidity and mortality.

## Figures and Tables

**Figure 1 diagnostics-15-02373-f001:**
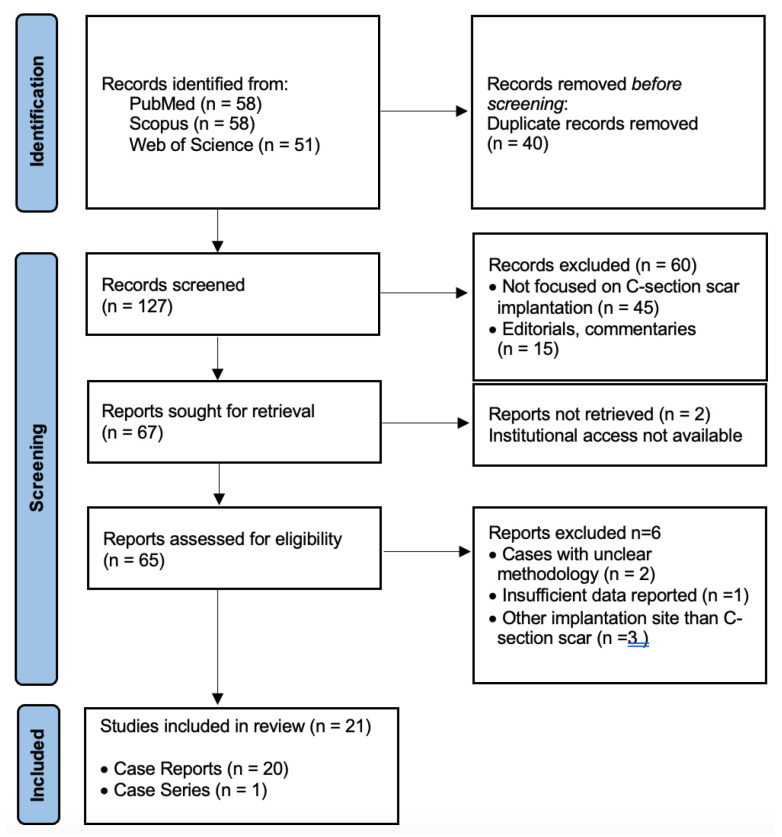
Prisma Flow Chart.

**Figure 2 diagnostics-15-02373-f002:**
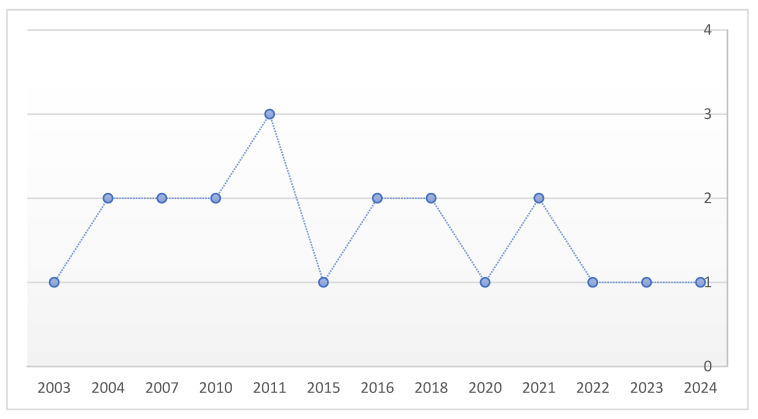
Chronological distribution of published HCSP cases between 2003 and 2024.

**Figure 3 diagnostics-15-02373-f003:**
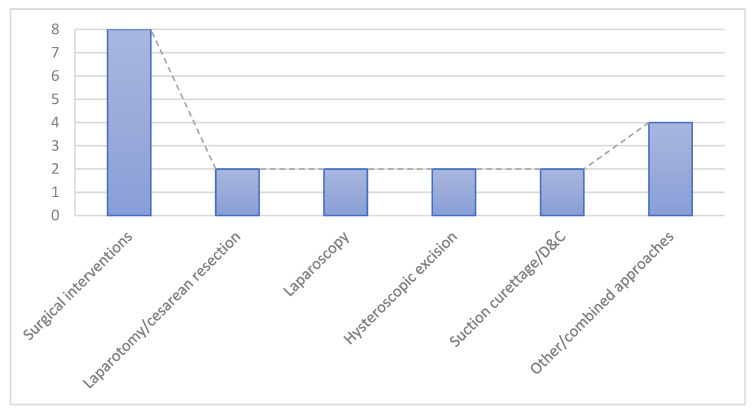
Distribution of management approaches among published HCSP cases.

## Data Availability

All data supporting the findings of this study are available within the article and its Appendix A.

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
