# Peer review of "Heterotopic Cesarean Scar Pregnancy: A Systematic Review of Diagnosis, Management and Prognosis"

_diagnostics, 2025, doi:10.3390/diagnostics15182373_

Round 1
Reviewer 1 Report
Comments and Suggestions for Authors
This systematic review is about heterotopic cesarean scar pregnancy (HCSP) an exceptionally rare and potentially life-threatening form of ectopic pregnancy. Its incidence has risen in recent years, primarily due to the increased rate of cesarean deliveries and the widespread use of assisted reproductive technologies. This is systematic review with aims to provide a comprehensive synthesis of published 26
evidence on HCSP, with a focus on epidemiology, diagnostic challenges, therapeutic strategies, complications, and maternal-fetal outcomes. A systematic literature search was conducted in PubMed, Scopus, and Web of Science up to May 2025, in accordance with PRISMA guidelines.
Very very interesting and rare systematic review, rare studies and manuscripts in this area. This pathology is not so common, but it should be thoughtful and all the more engines in the era of today's use of technologies for assisted fertilization as well as the frequency of birth to the cesarean section, literally cesarean section epidemic. Really the topic of the systematic article and the manuscript themselves are very meaningful and necessary to the scientific public, will complement the emphasis in science. So that the said article will be interesting to the wider reading audience of gynecologists, obstetricists, human reproductives and doctors of general medicine. Really very useful, interesting and usable data, I think this article should be accepted and published, as it will be very read and quoted.
The only competition in the entire manuscript is that the results and data should be presented more tabular and graphically, such a very vast and confusing. Prisma Chart is also needed.
Author Response
We sincerely thank you for the kind and encouraging words.
We fully agree that clear visual presentation of data is essential for accessibility and impact. Regarding the PRISMA flow chart, we would like to clarify that it was already included in Supplementary Material 2, but we acknowledge that its visibility could have been improved. In the revised version, we have now inserted the PRISMA diagram directly into the main text to ensure easier access for readers. In addition, we have addressed the request for more graphical presentation of data. Several figures have been added to summarize management strategies and temporal distribution of published cases.
Reviewer 2 Report
Comments and Suggestions for Authors
This manuscript addresses a very rare but clinically relevant entity, heterotopic cesarean scar pregnancy (HCSP). The systematic review is well-structured, follows PRISMA guidelines, and compiles an impressive number of reported cases. The topic is of high importance for obstetricians, gynecologists, and reproductive medicine specialists, given the rising rates of cesarean delivery and assisted reproductive technologies. The review provides valuable insights into diagnostic challenges, management strategies, and maternal–fetal outcomes.
That said, several points should be considered to strengthen the manuscript:
-
Methodology
-
The review is systematic, but the absence of PROSPERO registration or protocol publication should be acknowledged more explicitly as a limitation.
-
A formal risk-of-bias assessment was not performed; while understandable given the predominance of case reports, at least a structured discussion of potential biases (publication bias, reporting bias) would be helpful.
-
-
Results
-
The table is detailed and informative, but some synthesis through figures (e.g., flow chart of cases, management strategies by frequency, or timeline of reported cases) would greatly improve readability.
-
Clarify the handling of cases with incomplete outcome data (n=13). Were they excluded from quantitative analysis, or simply reported narratively?
-
-
Discussion
-
The discussion is comprehensive but somewhat repetitive. Condensing overlapping information and highlighting the most clinically relevant points would improve clarity.
-
Consider emphasizing the practical implications for clinicians: suggested diagnostic algorithms, warning signs for early detection, and considerations for counseling patients undergoing ART or with prior cesarean section.
-
Expand the section on long-term outcomes (future fertility, recurrence risk, uterine integrity) if such data are available in the literature.
-
-
References
-
References are appropriate and up to date. However, adding more recent reviews or guidelines on cesarean scar pregnancy would strengthen the introduction and discussion.
-
-
Limitations
-
The limitations of the evidence base (case reports, small series, lack of standardized management) should be highlighted more prominently in both the Discussion and Conclusions.
-
In summary, this review is a valuable contribution to the literature on HCSP. With some revisions to streamline the text, improve methodological transparency, and strengthen the discussion with practical implications, the manuscript will be suitable for publication.
Comments on the Quality of English LanguageThe overall level of English is acceptable and the manuscript is understandable. Nevertheless, the style would benefit from additional language editing to enhance clarity and fluency. Several sentences are overly long or contain redundancies, which may hinder readability for an international audience.
For example:
-
In the Introduction (lines 64–67), the sentence “Furthermore, the ectopic gestations may not be identifiable on either transvaginal or transrectal ultrasound, and serum β-hCG measurement has limited diagnostic reliability in HP, given its predominance by the intrauterine pregnancy” could be divided into shorter sentences to improve clarity.
-
In the Methods (lines 132–136), the phrasing “Given the rarity of HCSP and the predominance of case reports and small series, a formal risk-of-bias assessment was not feasible. Methodological quality was assessed descriptively using the CARE checklist for case reports and a narrative appraisal for the case series” could be streamlined by avoiding repetition of “case reports/series.”
-
In the Discussion (lines 355–364), the sentence “Multiple therapeutic approaches have been described in the literature, including laparotomic, laparoscopic or hysteroscopic resection, expectant management, suction curettage, or selective embryocidal interventions such as potassium chloride [KCl] injection. None of these methods have demonstrated clear superiority in terms of efficacy or safety” could be simplified by merging overlapping expressions.
Additionally, there are occasional shifts in terminology (e.g., “transvaginal ultrasonography,” “transvaginal sonography,” “TVUS,” “TVS”) that should be standardized to avoid confusion.
In summary, the English is adequate but could be significantly improved through minor language editing to shorten long sentences, reduce redundancy, and ensure consistency in terminology.
Author Response
We sincerely thank you for the careful evaluation of our manuscript and for the constructive suggestions provided. We have revised the manuscript accordingly. Below we provide a detailed, point-by-point response.
Comment 1: The absence of PROSPERO registration or protocol publication should be acknowledged more explicitly as a limitation.
Response 1: We thank the reviewer for this important observation. We have now acknowledged the absence of PROSPERO registration as a methodological limitation and clarified this both in the Methods section and in the Limitations subsection of the Discussion.
Comment 2: A formal risk-of-bias assessment was not performed; at least a structured discussion of potential biases would be helpful.
Response 2: We agree with this comment. While the predominance of case reports precluded a formal risk-of-bias analysis, we have expanded the Discussion to include a structured appraisal of potential biases (publication bias, reporting bias, and selective outcome reporting).
Comment 3: Figures would improve readability, including flow chart of cases, management strategies by frequency, or timeline of reported cases.
Response 3: We appreciate this suggestion and have revised the Results section accordingly. We added figures summarizing case distribution, management strategies, and temporal trends, which improve the visual clarity and accessibility of the findings.
Comment 4: Clarify the handling of incomplete outcome data (n=13).
Response 4: We thank the reviewer for pointing this out. We have clarified in the Methods and Results sections that these cases were included in the qualitative synthesis but excluded from quantitative outcome analyses.
Comment 5: The discussion is somewhat repetitive; condensing overlapping information and highlighting clinical relevance would improve clarity.
Response 5: We agree and have revised the Discussion to reduce redundancies and to emphasize key clinical implications, including diagnostic considerations, early warning signs, and counseling aspects for patients undergoing ART or with prior cesarean delivery.
Comment 6: Expand section on long-term outcomes (fertility, recurrence, uterine integrity).
Response 6: We have expanded the Discussion to address available data on future fertility, recurrence risk, and uterine integrity following HCSP treatment, supported by recently published studies.
Comment 7: Add more recent reviews or guidelines on cesarean scar pregnancy.
Response 7: We have updated the Introduction and Discussion with several recent studies and systematic reviews (2020–2025) identified during our search, to strengthen the context and support our findings.
Comment 8: Limitations of the evidence base should be highlighted more prominently.
Response 8: We agree and have expanded the Limitations section to emphasize the constraints of case reports and small series, the lack of standardized management, and the need for larger multicenter studies.
Comment 9: The English language requires some editing (overly long sentences, redundant phrasing, inconsistent terminology).
Response 9: We thank the reviewer for this remark. The manuscript has undergone thorough language revision to improve clarity and conciseness. Long sentences were shortened, redundancies eliminated, and terminology standardized (e.g., consistently using “transvaginal ultrasonography [TVUS]”).